# Antimicrobial Resistance and Extended-Spectrum Beta-Lactamase Genes in *Enterobacterales*, *Pseudomonas* and *Acinetobacter* Isolates from the Uterus of Healthy Mares

**DOI:** 10.3390/pathogens12091145

**Published:** 2023-09-08

**Authors:** Pamela Thomson, Patricia García, Camila del Río, Rodrigo Castro, Andrea Núñez, Carolina Miranda

**Affiliations:** 1Laboratorio de Microbiología Clínica y Microbioma, Escuela de Medicina Veterinaria, Facultad de Ciencias de la Vida, Universidad Andrés Bello, Santiago 8370134, Chile; mcm@unab.cl; 2Departamento de Laboratorios Clínicos, Escuela de Medicina, Pontificia Universidad Católica, Santiago 8940000, Chile; pgarciacan@uc.cl; 3Escuela de Medicina Veterinaria, Facultad de Recursos Naturales y Medicina Veterinaria, Universidad Santo Tomás, Talca 3473620, Chile; 4Escuela de Medicina Veterinaria, Facultad de Ciencias Agrarias y Forestales, Universidad Católica del Maule, Curicó 3340000, Chile; 5Facultad de Medicina Veterinaria y Agronomía, Universidad de las Américas, Santiago 7500975, Chile; 6Laboratorio de Microbiología Red de Salud UC-CHRISTUS, Pontificia Universidad Católica, Santiago 8940000, Chile; cmirandt@uc.cl

**Keywords:** uterus, mares, antibiotic resistance, ESBL

## Abstract

Antibiotic-resistant bacteria are a growing concern for human and animal health. The objective of this study was to determine the antimicrobial resistance and extended-spectrum beta-lactamase genes in *Enterobacterales*, *Pseudomonas* spp. and *Acinetobacter* spp. isolates from the uterus of healthy mares. For this purpose, 21 mares were swabbed for samples, which were later seeded on blood agar and MacConkey agar. The isolates were identified using MALDI-TOF and the antimicrobial susceptibility test was performed using the Kirby–Bauer technique. To characterize the resistance genes, a polymerase chain reaction (PCR) scheme was performed. Of the isolates identified as Gram-negative, 68.8% were *Enterobacterales*, represented by *E. coli*, *Enterobacter cloacae*, *Citrobacter* spp., and *Klebsiella pneumoniae*; 28.1% belonged to the genus *Acinetobacter* spp.; and 3.1% to *Pseudomonas aeruginosa*. A 9.3% of the isolates were multidrug-resistant (MDR), presenting resistance to antibiotics from three different classes, while 18.8% presented resistance to two or more classes of different antibiotics. The diversity of three genes that code for ESBL (*bla*_TEM_, *bla*_CTX-M_ and *bla*_SHV_) was detected in 12.5% of the strains. The most frequent was *bla*_SHV_, while *bla*_TEM_ and *bla*_CTX-M_ were present in *Citrobacter* spp. and *Klebsiella pneumoniae*. These results are an alarm call for veterinarians and their environment and suggest taking measures to prevent the spread of these microorganisms.

## 1. Introduction

The presence of microorganisms in the uterus of mares without reproductive pathologies has shown the existence of a uterine microbiota, of which nearly 200 microorganisms have been identified by molecular techniques [1,2,3,4,5]. These microorganisms play fundamental roles [6] in processes such as embryo implantation, prevention of the growth of pathogenic microorganisms, and protection of the epithelium [7,8,9]. It has been described that alteration or dysbiosis is related to the direct entry of bacteria through mating, artificial insemination, gynecological examination [1,6,10,11], malformation of the vulva or perineal region [12,13], and lesions of the cervix or vagina. These situations have been positively associated with bacterial endometritis [9,14], where *Streptococcus equi*, *Escherichia coli*, *Klebsiella* spp., and *Pseudomonas* spp. are the most frequently isolated microorganisms in this pathology [15,16,17,18,19].

In some countries in Europe, India, and the United States, bacteria such as *Enterobacterales*, *Pseudomonas* spp., and *Acinetobacter* spp. isolated from the uterus and vagina of healthy mares [11,16,18,20,21,22,23] show resistance or multidrug resistance (MDR) to antibiotics [24,25,26,27,28,29,30,31]. Extended-spectrum beta-lactamase (ESBL)-producing *Enterobacteriaceae* and carbapenem-resistant *Acinetobacter baumannii* and *Pseudomonas aeruginosa* were classified by the World Health Organization (WHO) as critically important pathogens [32] and are among the main antimicrobial resistance (RAM) threats in humans [33,34] and animals [35,36,37,38,39]. These situations are a cause of concern for the medical environment due to the probable dissemination of microorganisms and the limitation of therapeutic options [40]. The objective of this study was to determine the antimicrobial resistance and extended-spectrum beta-lactamase genes in *Enterobacterales*, *Pseudomonas* spp., and *Acinetobacter* spp. isolates from the uterus of healthy mares.

## 2. Materials and Methods

### 2.1. Ethical Approval

This research was approved by the Scientific Committee of Ethics of the Central-South macrozone of the Santo Tomás University, Chile (Approval number 60-21) and was carried out in the Maule region (35°25′ S, 71°39′ W) during October 2021.

### 2.2. Subjects and Criteria Selection

The group consisted of 21 purebred Chilean mares, between 4 and 15 years old, who were fed in a mixed meadow of ryegrass and white clover with free access to water. All participants were off antibiotic treatment for at least one month before being sampled

Clinically healthy mares in the ovulatory phase were included; this was determined by a gynecological clinical examination, transrectal ultrasound (Chison Eco 6 ultrasound, 5 MHz linear transducer), and cytology [13]. No mare presented a record of abortion, embryonic losses, endometritis, dystocia, or any reproductive pathology.

### 2.3. Uterine Samples, Collection, Isolation, and Identification

To avoid contamination, the tail was covered with sterile gauze and the fecal material was removed from the rectum. Subsequently, the perineum, clitoral fossa, and vulva were washed with soap and lukewarm water. The vulva was dried with bleached paper and the procedure was repeated until the area was visibly clean. Sample contamination was minimized using a sterile rectal glove, and a double-guarded occluded swab (IMV, Legler, Limoges, France) [18].

The uterine swabs were introduced directly in Amies transport medium (Linsan, Santiago, Chile) and were immediately transferred to the Clinic Microbiology and Microbiome laboratory, where they were seeded within 24 h on blood agar and MacConkey agar (Merk, Darmstadt, Germany). All plates were incubated at 37 °C for 18 to 24 h. Semi-quantitative evaluation of the different morphotypes was carried out, so that those that showed abundant growth in the second quadrant of the clock were selected with the help of a magnifying glass, observing standard patterns such as colony shape, borders, topography, color, and texture [41]. Later, each morphotype was isolated on blood agar (Linsan, Santiago, Chile) and was identified using matrix-assisted laser desorption/ionization time-of-flight (MALDI-TOF) mass spectrometry analysis (MALDI Biotyper, Bruker, Singapore) following the manufacturer’s instructions and as described previously [42,43]. Importantly, *Citrobacter* spp. were analyzed as members of the *C. freundii* complex due to the impossibility of identification down to the species level using the MALDI-TOF technique [44].

### 2.4. Antimicrobial Susceptibility Testing

All isolates corresponding to *Enterobacterales*, *Pseudomonas* spp., and *Acinetobacter* spp. were tested against a panel of 13 antibiotics using the disk diffusion Kirby–Bauer method following CLSI guidelines in the M100 and VET01S [45,46]. Antibiotics tested included amikacin (AMK, 30 μg); ampicillin (AMP, 10 μg); amoxicillin/clavulanic acid (AUG 20/10 μg); ceftazidime (CAZ, 30 μg); ciprofloxacin (CIP, 5 μg); ceftriaxone (CRO, 30 μg); doxycycline (DXT, 30 μg); enrofloxacin (ENR, 5 μg); ertapenem (ETP, 10 μg); gentamicin (GEN, 10 μg); imipenem (IPM, 10 μg); ampicillin/sulbactam (SAM, 10/10 μg); and trimethoprim/sulfamethoxazole (SXT, 1.25/23.75 μg). All of these were supplied by OXOID (Hampshire, UK). The AMP disc was not used for *Pseudomonas* spp., *Enterobacter* spp., *Citrobacter* spp., and *Klebsiella* spp. In addition, for *Acinetobacter* spp. we only tested carbapenems, within the group of beta-lactams [46].

The screening of organisms producing extended-spectrum b-lactamases (ESBLs) and/or AmpC was performed using Cefotaxime 30 μg (CTX), CTX + clavulanic acid, and CTX + cloxacillin disc (Liofilchem, Teramo, Italy). For *Acinetobacter* spp. and *P. aeruginosa*, CTX + clavulanic acid + cloxacillin disc (Liofilchem, Teramo, Italy) were used to inhibit the chromosomal AmpC b-lactamase, which can antagonize the synergistic effect with clavulanate [46].

In all experiments, *Klebsiella quasipneumoniae* ATCC 700603 and *E. coli* ATCC 25922 were used as resistant and susceptible controls, respectively. Bacterial isolates resistant to ≥1 agent in >3 antimicrobial different classes were cataloged as multidrug-resistant (MDR) following previously standardized criteria [47].

The genes encoding ESBLs (*bla*_CTX-M_, *bla*_SHV_, *bla*_TEM_, *bla*_PER_ and *bla*_GES_) were detected by a conventional PCR scheme, as previously reported [30,48,49]. Briefly, PCR was performed in a volume of 25 µL in a Veriti^®^ thermal cycler (Applied Biosystems). The reaction mix contained 1× Green GoTaq^®^Flexi Buffer (Promega, Madison, WI, USA), PCR buffer, 800 nM of each primer (Table 1), 200 nM of dNTPs, 1.5 nM of MgCl2, 5 µL of DNA, and 1 U of Taq polymerase. The amplification program was an initial denaturation of 5 min at 94 °C, then 35 equal cycles of 40 s at 94 °C, 40 s at 52/57 °C, and 60 s at 72 °C. Finally, a final extension of 7 min at 72 °C occurred (Table 1). Tubes were stored at 4 °C until detection of the PCR product on 1.5% agarose gel. In cases where there was doubt that the size obtained in the PCR product was a non-specific amplification, it was decided to sequence them to confirm that they corresponded. For this, they were sequenced by the Sanger method using the BigDye^®^ Terminator v3.1 Cycle Sequencing Kit (Applied Biosystems, Waltham, MA, USA) in the SeqStudio Genetic Analyzer (Applied Biosystems). The obtained sequences were compared with the GenBank database using the BLAST program. The sequences were deposited in the NCBI GenBank database (accession numbers OR242737, OR242738, OR242739, OR242740, and OR242741).

### 2.5. Data Analysis

Data analysis was performed using Excel (Microsoft 365), open-source statistical computing package release 1.3.4; upset plots [50] were made employing the UpSetPlot package (https://github.com/jnothman/UpSetPlot, access on 25 April 2022), release 0.6.0.

## 3. Results

The group of 21 mares studied were composed of purebred Chilean females, with more than two births, between 4 and 15 years old, with an average age of 8 years.

As a result, each mare had a mixed bacterial growth in 100% of the cases, isolating at least two or three different bacteria, corresponding to 42.8% and 52.4%, respectively. In contrast, the largest bacterial population with four isolates was present in only one mare, representing 4.8%.

Of the 55 bacterial isolates, 39 (70.9%) were Gram-negative bacteria of which 22 (56.4%) belonged to *Enterobacterales*, represented by 17 *Escherichia coli*, 3 *Enterobacter cloacae*, 1 *Citrobacter* spp., and 1 *Klebsiella pneumoniae*; and 17 were Gram-negative non-fermenting rod bacteria, represented by 9 *Acinetobacter* spp. (2 *A. lwoffii*, 1 *A. johnsonii*, and 6 *Acinetobacter* spp.), 3 *Brevundimonas diminuta*, 3 *Sphingobacterium* spp., 1 *Pseudomonas aeruginosa*, and 1 *Ochrobactrum anthropi.* In contrast, 16 (29.1%) of the isolates were Gram-positive bacteria: 10 *Enterococcus* spp. (6 *E. casseliflavus*, 2 *E. faecalis*, and 2 *E. faecium*), 4 *Streptococcus equi*, and 1 *Corynebaterium* sp. (Figure 1).

Of all antibiotics tested against the *Enterobacterales* and *Pseudomonas* spp. (*n* = 23), the third generation cephalosporins were shown a 26.1% resistance, followed by GM. In the strains of the *Acinetobacter* spp. tested, AMP was shown the greatest resistance with 44.4% (Figure 2). Only 9.3% (*n* = 32) of isolates showed MDR, corresponding to *Escherichia coli*, *Enterobacter cloacae*, and *Klebsiella pneumoniae*; also, 18.8% of isolates presented resistance to two or more classes of antibiotics. On the contrary, the carbapenems IPM and ETP were sensitive to all isolates (Table 2).

These results revealed the diversity of three ESBL genes (Table 3), *bla*_TEM_, *bla*_CTX-M_ and *bla*_SHV_, in four isolates found (*Escherichia coli*, *Acinetobacter* spp., *Klebsiella pneumoniae* and *Citrobacter* spp.). The most frequent was *bla*_SHV_, being detected in all four isolates, while *bla*_TEM_ and *bla*_CTX-M_ were only present in *Klebsiella pneumoniae and Citrobacter* spp., which also exhibited coexistence of the three genes (*bla*_CTX-M_, *bla*_SHV_ and *bla*_TEM_).

## 4. Discussion

It is currently recognized that mares and their reproductive environment are a source of origin of different microorganisms, which can be disseminated to other animals and to humans through direct and indirect contact [40,51].

Of the group of mares sampled, all presented between two and four different bacterial isolates, information which is consistent with previous articles [4,11,12,13,14,15,16,17,18,19,20,21,22,23,52,53]. According to these results, the *Enterobacterales* group was the most predominant, with *Escherichia coli* being the most frequent bacterium [17,54,55,56]. Some studies indicate that the source of origin of this agent would be fecal matter that contaminates the vulva and perineum, associated with a bad anatomical conformation [12,13,15]. Likewise, *Enterococcus* spp., *Acinetobacter* spp., *Streptococcus equi* spp. and to a lesser extent *Klebsiella pneumoniae* and *Pseudomonas aeruginosa* [4,53,56,57,58,59,60,61]. 

A retrospective study carried out with bacteria isolated from the reproductive tract of 4122 mares with endometritis identified *Escherichia coli* and *Streptococcus equi* more frequently, adding that over the years the antimicrobial efficacy of cefquinome against *E. coli* decreased significantly, and the same is true of ampicillin, cefquinome, and penicillin against *S. equi* [22]. Previously, *Pseudomonas* spp. were reported in mares with fertility problems [62] in chronic endometritis and in those without response to antibiotic treatments [63,64]. On the other hand, *Klebsiella pneumoniae* has been frequently found in respiratory and digestive conditions in horses [65,66]. *Enterococcus* spp. are rare to find in the uterus of healthy mares; however, *E. faecalis* has been isolated from mares with chronic endometritis and infertility [67]. *Acinetobacter baumannii* has been found in the uterus of healthy mares, and in equine patients with wounds, septicemia, eye infections, bronchopneumonia, neonatal encephalopathy, and venous catheter [36,37,68,69]. Therefore, Lupo et al. indicate that *A. baumannii* has become a nosocomial pathogen in veterinary hospitals [70]. Finally, *Citrobacter freundi*, along with other species, has been detected in a case of uterine infection with purulent discharge [71] and another of endocarditis in a foal [72].

Although antimicrobial resistance (RAM) has been previously reported in bacteria from the equine reproductive tract [11,22,62,73], it is unknown whether its origin is related to local or systemic treatments. Similarly, it is considered that exposure to low levels of antibiotics through semen diluents would be a probable cause of antibiotic resistance in mares [62,73]. RAM has been identified as one of the major problems facing human and animal health [74,75,76]; in this regard, the British Equine Veterinary Association has indicated that the use of fluoroquinolones and third generation cephalosporins should be regulated in empirical or prophylactic therapies [77,78]. In addition, Benko et al., point out that microorganisms such as *Pseudomonas*, *Klebsiella*, and *Escherichia coli* are considered highly resistant to β-lactam class antibiotics and ampicillin [16].

These results exhibited high resistance to the third generation cephalosporins and ampicillin, which coincides with what was mentioned in other studies [79,80,81,82] and could be attributed to the empirical use of ceftiofur in equine Gram-negative bacterial infections [78,83,84,85,86]. Likewise, the use of broad-spectrum cephalosporins could be a selection factor for bacteria with ESBL. Generally, the resistance genes that code for the expression of this ESBL phenotype, such as *bla*_CTX-M,_
*bla*_SHV_, *bla*_TEM_, and *bla*_CTX-M_, are in plasmids or integron-type structures, which would facilitate horizontal gene transfer [30,40,66,87,88,89,90].

In 1998, the first ESBL strain was detected in bacteria of animal origin, an *E. coli* carrier of the SHV-12 gene [91]. Currently, there are several authors who report the presence of these bacteria in different animal species, emphasizing transmission to humans [26,28,29,87,92,93,94,95,96,97]. A study carried out from rectal swab samples in pairs of healthy mares and newborn foals reported the presence of ESBL in *Escherichia coli* strains in 25% and 29%, respectively, noting that during hospitalization this number increased significantly [98], the same as previously reported [99], suggesting an association with high use of antimicrobials, even in untreated animals [26], and nosocomial acquisition of ESBL in a hospital setting [100,101]. On the other hand, *Klebsiella* spp. have become a major health problem, leading to treatment failure in humans and animals. A previous report, carried out in healthy horses, reported only one *K. pneumoniae* isolate confirmed as a producer of ESBL (*bla*_CTX-M_) [92]. Another study carried out in the USA recovered *E. coli* isolates from clinical samples of equine patients over a period of five years, finding resistance to ceftiofur in 13 out of 48 of them, while the CTX-M and SHV genes were detected in 7 of them, which code for ESBL [84].

Interestingly, the literature reports that bacteria such as *Acinetobacter* spp., *E. coli*, *Klebsiella* spp., and *Pseudomonas* spp. can survive on inanimate surfaces for months [102,103]. The persistence of these bacteria ranges from 3 days to 5 months for *Acinetobacter* spp. [104], 1.5 h to 16 months for *E. coli*, 2 h to 30 months for *Klebsiella* spp., and 6 h to 16 months for *Pseudomonas* spp. [84,105,106]. The emergence of ESBL-producing bacteria is alarming, necessitating surveillance studies to understand the transmission and epidemiology of such microorganisms [30,84]. It is necessary to consider reinforcing measures such as hygiene, hand washing, identification of infected patients, cleaning, and disinfection of environmental sources of contamination.

ESBL-producing *Enterobacteriaceae* are a global public health alert, and so antimicrobial efficacy monitoring programs are crucial to consciously use antibiotics and preserve their effectiveness for both human and veterinary medicine.

The contribution of this study is its focus on antimicrobial resistance. So far, there is no evidence related to the presence of ESBL in isolates of *Enterobacterales*, *Pseudomonas*, and *Acinetobacter* obtained from a population of reproductively active mares in Chile. Although the group of animals studied was limited, the results revealed the diversity of three ESBL genes, *bla*_TEM_, *bla*_CTX-M_ and *bla*_SHV_, which co-existed in *Klebsiella pneumoniae* and *Citrobacter* spp. It would be interesting to continue with these studies and apply them to a larger population to include other molecular techniques for the detection of resistance genes, such as the sequencing of the complete genome, to better understand the situation at the national level. This communication alerts us to the presence of multi-resistant bacteria in the uterus of healthy mares and urges us to emphasize cleaning and disinfection protocols to prevent dissemination at the animal–human–environment interface.

## Figures and Tables

**Figure 1 pathogens-12-01145-f001:**
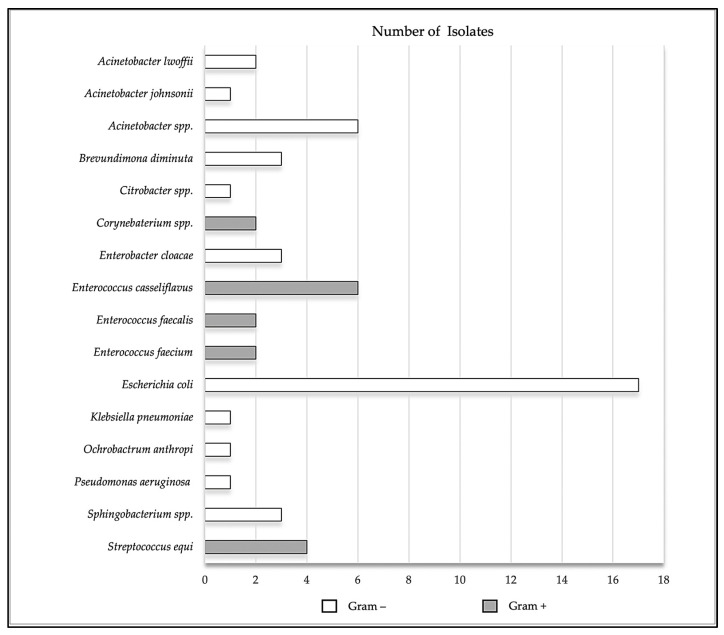
Total number of isolates present in the uterus of healthy mares, represented by *Escherichia coli* (17), *Enterobacter cloacae* (3), *Citrobacter* spp. (1), *Klebsiella pneumoniae* (1), *Acinetobacter* spp. (9), *Pseudomonas aeruginosa* (1), *Brevundimonas diminuta* (3), *Sphingobacterium* spp. (3), *Ochrobactrum anthropi* (1), *E. casseliflavus* (6), *E. faecalis* (2), *E. faecium* (2), *Enterococcus* spp. (10), *Streptococcus equi* (4) and *Corynebaterium* spp. (2).

**Figure 2 pathogens-12-01145-f002:**
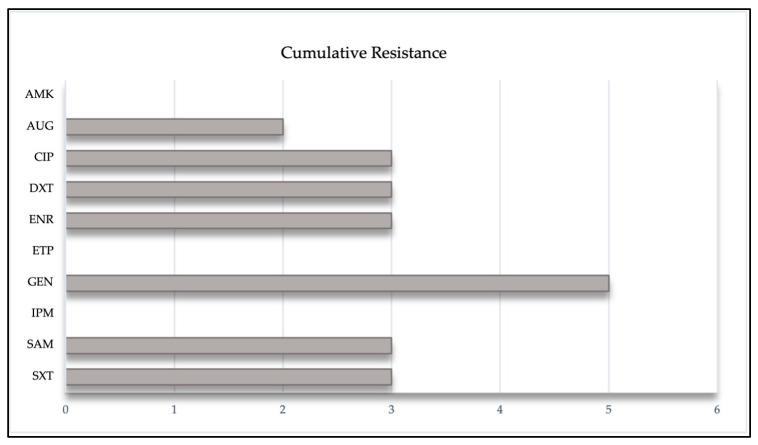
Representation of the cumulative resistance of different antibiotics against *Enterobacterales*, *Acinetobacter* spp. and *Pseudomonas aeruginosa*, isolated from the uterus of mares.

**Table 1 pathogens-12-01145-t001:** Primers used for ESBL PCRs.

β-Lactamase Gene	Primer Sequence (5′ → 3′)	Product Length (bp)	T° Annealing
*bla* _CTX-M_	MA1: SCSATGTGCAGYACCAGTAA	554	57 °C
	MA2: CCGCRATATGRTTGGTGGTG		
*bla* _SHV_	SHV F: TCGGCCTTCACTCAAGGATG	785	57 °C
	SHV R: ATGCCGCCGCCAGTCATATC		
*bla* _TEM_	TEM F: TTAGACGTCAGGTGGCACTT	972	52 °C
	TEM R: GGACCGGAGTTACCAATGCT		
*bla* _PER_	PER F: AAAGAGCAAATTGAATCCATAGTC	835	57 °C
	PER R: GTTAATTTGGGCTTAGGGCAG		
*bla* _GES_	GES F: ATGCGCTTCATTCACGCAC	864	57 °C
	GES R: CTATTTGTCCGTGCTCAGG		

**Table 2 pathogens-12-01145-t002:** Specific antibiotic resistance against 32 isolates: *Enterobacterales*, *Acinetobacter* spp. and *Pseudomonas* spp.

Isolate	Specific Antibiotic Resistance
*Acinetobacter lwoffii*	AMP
*Acinetobacter lwoffii*	-
*Acinetobacter* spp.	-
*Acinetobacter* spp.	AMP
*Acinetobacter* spp.	AMP
*Acinetobacter* spp.	AMP
*Acinetobacter* spp.	-
*Acinetobacter* spp.	-
*Acinetobacter johnsonii*	-
*Citrobacter* spp.	CAZ-SAM-CRO
*Enterobacter cloacae*	CAZ-GM-CRO
*Enterobacter cloacae*	SAM-CIP-SXT-GM-CRO-ENR
*Enterobacter* spp.	CAZ-GM
*Escherichia coli*	CAZ-CIP-SXT-GM-CRO-AMP-ENR-AUG-DXT
*Escherichia coli*	DXT
*Escherichia coli*	-
*Escherichia coli*	-
*Escherichia coli*	-
*Escherichia coli*	-
*Escherichia coli*	-
*Escherichia coli*	-
*Escherichia coli*	-
*Escherichia coli*	-
*Escherichia coli*	-
*Escherichia coli*	-
*Escherichia coli*	-
*Escherichia coli*	-
*Escherichia coli*	-
*Escherichia coli*	-
*Escherichia coli*	-
*Klebsiella pneumoniae*	CAZ-SAM-CIP-SXT-GM-CRO-ENR-AUG-DXT
*Pseudomonas* spp.	-

**Table 3 pathogens-12-01145-t003:** Genetic characterization of extended-spectrum beta-lactamase (ESBL)-positive bacterial isolates of equine origin.

Species	ESBL	*bla* Genes
*Acinetobacter lwoffii*	−	−
*Acinetobacter* spp.	+	*bla* _SHV_
*Acinetobacter* spp.	+	−
*Acinetobacter* spp.	+	−
*Citrobacter* sp.	+	*bla*_CTX-M_, *bla*_TEM_, *bla*_SHV_
*Enterobacter cloacae*	−	−
*Enterobacter cloacae*	−	−
*Escherichia coli*	+	*bla* _SHV_
*Klebsiella pneumoniae*	+	*bla*_CTX-M_, *bla*_TEM_, *bla*_SHV_

## Data Availability

Not applicable.

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
