# Peer review of "Antimicrobial Resistance and Extended-Spectrum Beta-Lactamase Genes in Enterobacterales, Pseudomonas and Acinetobacter Isolates from the Uterus of Healthy Mares"

_pathogens, 2023, doi:10.3390/pathogens12091145_

Round 1

Reviewer 1 Report

Dear sir,

We usually do NOT test pseudomonas, enterobacter, citrobacter and klebsiella for ampicillin resistance, because they are intrinsic resistant. You have to erase it from methods, results and tables.

For the same reason, we do NOT test acinetobacter for beta-lactams resistance other than carbapenems. You have to erase it.

Author Response

Comments and Suggestions for Authors

Dear sir,

We usually do NOT test pseudomonas, enterobacter, citrobacter and klebsiella for ampicillin resistance, because they are intrinsic resistant. You have to erase it from methods, results and tables.

For the same reason, we do NOT test acinetobacter for beta-lactams resistance other than carbapenems. You have to erase it.

RESPONSE

Dear Sir

Thanks for your contributions. All his suggestions in the text were applied

Reviewer 2 Report

The manuscript titled "Antimicrobial resistance and Extended-Spectrum Beta-Lactamase genes in Enterobacterales, Pseudomonas, and Acinetobacter isolates from the uterus of healthy mares" is a good piece of work. However, there are many concerns that should be addressed for better clarity and understanding.

In the abstract, the sentence "A 9.3% of the isolates were multidrug resistance (MDR) corresponding, while 32.3% presented resistance to two classes of antibiotics" needs to be rephrased to ensure the meaning is clear.

Additionally, the term MDR is frequently used throughout the manuscript. It is important to explain how the isolates were categorized as MDR and provide the criteria that were followed.

In the discussion, the authors have used the abbreviation "antibiotic resistance (RAM)". However, this is not the standard abbreviation, and it would be beneficial to use the appropriate term or abbreviation.

Figure 2 lacks clarity regarding what the authors intend to convey. It would be helpful to provide a more explicit explanation or description of the figure's purpose.

Regarding the ESBL genotypes, it would be valuable to discuss whether they corresponded with the phenotype results in the discussion section. This would enhance the understanding of the findings and their implications.

The quality of the English in the manuscript is poor and requires thorough editing

Author Response

Thanks for your contributions, below you will see the response of each one

In the abstract, the sentence "A 9.3% of the isolates were multidrug resistance (MDR) corresponding, while 32.3% presented resistance to two classes of antibiotics" needs to be rephrased to ensure the meaning is clear.

  • The sentence was changed

Additionally, the term MDR is frequently used throughout the manuscript. It is important to explain how the isolates were categorized as MDR and provide the criteria that were followed.

                -The term MDR is defined in M & M. Even so, it was added in the abstract

In the discussion, the authors have used the abbreviation "antibiotic resistance (RAM)". However, this is not the standard abbreviation, and it would be beneficial to use the appropriate term or abbreviation.

  • The abbreviation was corrected

Figure 2 lacks clarity regarding what the authors intend to convey. It would be helpful to provide a more explicit explanation or description of the figure's purpose.

                - Figure 2 was changed to a table that better explains the results.

Regarding the ESBL genotypes, it would be valuable to discuss whether they corresponded with the phenotype results in the discussion section. This would enhance the understanding of the findings and their implications.

                - The ESBL phenotype of the studied strains was evaluated again and expressed in results.

Round 2

Reviewer 2 Report

The suggested changes have been incorporated therefore can be accepted. 

The English language in the manuscript is acceptable. However, there were minor grammatical and syntactic mistakes that need to be corrected.

Author Response

Santiago, 28 August 23
Dear reviewer, according to his request we have revised the English language again throughout the text and we appreciate your contributions.
Cordially